# YOLOv3_ReSAM: A Small-Target Detection Method

**Bailin Liu [1,2,*], Huan Luo [2,3], Haotong Wang [1,2] and Shaoxu Wang [3]**

[1] School of Computer Science and Engineering, Xi'an Technological University, Xi'an 710021, China; wanghaotong@st.xatu.edu.cn

[2] New Network and Testing Control National and Local Joint Engineering Laboratory, Xi'an 710021, China; luohuan@st.xatu.edu.cn

[3] Ordnance Science and Technology College, Xi'an Technological University, Xi'an 710021, China; wangshaoxu@st.xatu.edu.cn

[*] Correspondence: xatulbl@xatu.edu.cn

**Abstract:** Small targets in long-distance aerial photography have the problems of small size and blurry appearance, and traditional object detection algorithms face great challenges in the field of small-object detection. With the collection of massive data in the information age, traditional object detection algorithms have been gradually replaced by deep learning algorithms and have an advantage. In this paper, the YOLOV3-Tiny backbone network is augmented by using the pyramid structure of image features to achieve multi-level feature fusion prediction. In order to eliminate the loss of spatial feature information and hierarchical information caused by pooling operations in convolution processes and multi-scale operations in multi-layer structures, a spatial attention mechanism based on residual structure is proposed. At the same time, the idea of reinforcement learning is introduced to guide bounding box regression on the basis of the rough positioning of the native boundary regression strategy, and the variable IoU calculation method is used as the evaluation index of the reward function, and the boundary regression model based on the reward mechanism is proposed for fine adjustment. The VisDrone2019 data set was selected as the experimental data support. Experimental results show that the mAP value of the improved small-object detection model is 33.15%, which is 11.07% higher than that of the native network model, and the boundary regression accuracy is improved by 23.74%.

**Keywords:** small target; YOLOv3; residual structure; spatial attention; boundary regression

## 1. Introduction

With the advantage of having a large perspective at high altitude, security patrol UAVs can eliminate blind spots in human patrols, quickly and efficiently extract feature information, automatically identify and locate targets in images, quickly capture suspicious non-cooperative targets and use them to accurately locate and identify targets in border patrol and border reconnaissance, saving a lot of manpower and time costs [1]. However, long-distance shooting images are different from still images in natural scenes, with high background complexity, small target size and blurry appearance, and target detection technology based on traditional methods has low accuracy for small-target detection and weak adaptability to targets in complex backgrounds, which are prone to leakage of alarms and false alarms, so the aerial small-target detection technology applied to low-altitude UAV platforms has important research significance. Traditional methods usually use server-side processing and send processing results to the on-board terminal, in which a large number of network requests consume a lot of computer resources while maintaining good network conditions. With the rise of artificial intelligence technology, convolutional neural network models are constantly changing, and models are constantly optimizing parameter redundancy, which can be directly deployed on airborne devices with limited computing resources. Therefore, under the premise of ensuring the correct rate, how to

greatly reduce the number of parameters and calculations of the network model has practical application value.

## 2. Related Work

Aiming at the problem that the target scale in the UAV aerial image changes greatly and the proportion of small targets is much higher than that in the natural scene image [2,3], the convolutional neural network is effectively integrated into the target detection algorithm of aerial images [4], but the early convolutional neural network model is complex and the target detection efficiency is low. By constantly adjusting the network structure, the size of the anchor frame and the resolution of the feature map, it can adapt to small-target detection [5]. Although the convolutional neural network has a certain generalization ability, the convolution and pooling operations in the network make it difficult to detect small targets. Yoshihashi et al. in [6] proposed a joint framework that uses spatio-temporal information to detect and track small flying objects simultaneously. Their deep learning network consisted of four modules: convolutional, ConvLSTM, intercorrelation and fully connected layer. Learning motion patterns from multi-frame images with ConvLSTM improves inspection performance; however, these networks are computationally very expensive [7]. To achieve feature extraction at different scales and improve the performance of small-target detection, the bottom features and top features with rich semantic information are fused [8], but for targets of different sizes, the generalization learning ability is low. For this optimization structure, the effect of small-target detection is improved to a certain extent by using horizontal connection [9,10] to strengthen feature propagation.

Inspired by the human visual system, many researchers have introduced attention modules into convolutional neural networks to improve target detection performance in recent years. Attention modules are mainly divided into channel attention [11], spatial attention [12] and channel and spatial mixed attention [13]. Because the imaging angle of low-altitude UAV aerial photography is different from that of natural scene images, small targets usually exist in long-distance images, and the backgrounds are complex. The unique shooting angle and larger width also bring noise interference to the detection task. Therefore, the spatial attention module [14–16] is usually introduced into the target detection task to weight the characteristics of the target area, and the feature extraction network can selectively focus on the target area containing important information, suppress other irrelevant information and reduce the influence of background on the detection results so as to improve the detection performance of the model for small targets. The attention module is introduced to gradually extract the target region through the cascaded attention model and generate the convolution features of attention perception to guide the feature learning process, highlight the features related to the target and suppress the interference of the background [17].

While improving the accuracy of target detection, researchers are also constantly exploring to improve the regression efficiency of target bounding boxes and the positioning accuracy of bounding boxes. The YOLO detection framework [18] divides the input image into grids of different scales and then predicts two frames for each grid, each frame containing the confidence of the target and the probability of the target category and finally removes the less likely target window and redundant window to obtain the final detection result. YOLO introduces the regression idea to transform the target detection problem into a regression problem, which can greatly improve the detection speed. However, because it only uses grids with different scales for regression, it cannot accurately locate the target, so the accuracy is reduced. With the wide application of deep reinforcement learning, a target location algorithm based on deep reinforcement learning can be used [19]. The agent can perform a series of simple deformation actions on the bounding box according to the learned strategy by regarding the entire picture as an environment, and an agent is introduced to learn a top-down search strategy for the bounding box. The agent can perform a series of simple deformation actions on the bounding box according to the learned strategy, and the deep reinforcement learning algorithm based on multi-agent cooperation

is used to learn the optimal strategy of cooperative target location [20]. The algorithm can effectively improve target location accuracy and locate the target accurately according to this contextual information.

## 3. Materials and Methods

This paper counts the use of the target detection algorithm represented by YOLO and the target algorithm represented by RCNN in the VisDrone 2019 Challenge, of which the target detection algorithm represented by YOLO has a small amount of operation and occupies an advantage in real-time detection, but the detection accuracy is low, and poor use only accounts for 4% of the use of the entire participating team, but this article considers its lightweight characteristics while taking into account the performance, so the YOLO series model is selected as the backbone model. Among them, the YOLOv3-Tiny network model inherits the superior performance advantages of the YOLOv3 series model in end-to-end real-time object detection and has the advantages of both being lightweight and real-time detection. However, there are still deficiencies in detection performance, and there is a semantic missing problem for the extraction of fine-grained target features which needs to be adjusted and optimized by the model structure to make up for the lack of detection performance. In this paper, the structure of the backbone network is modified as shown in Figure 1.

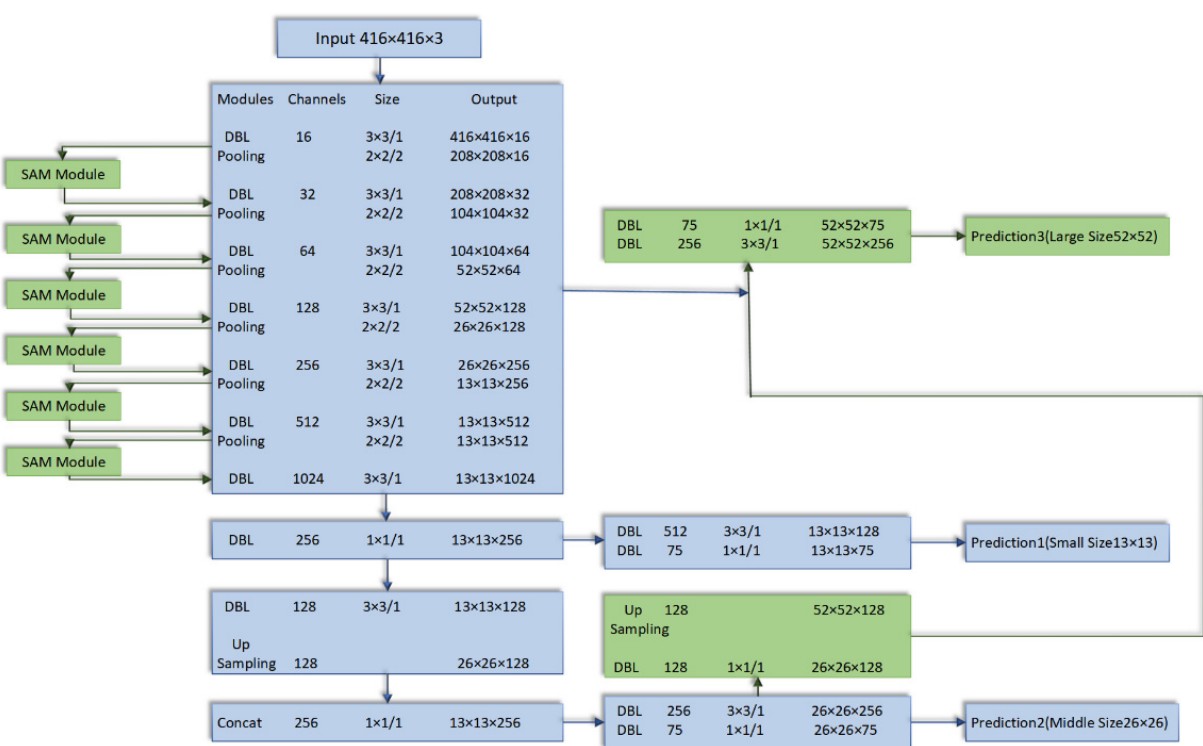

**Figure 1.** YOLOv3-Tiny_ReSAM network structure (the green box is the improvement part).

### 3.1. Expand the Image Feature Pyramid Structure

The common target detection network model uses top-level features for prediction, as shown in Figure 2. The semantic information of top-level features is rich, but the fine-grained features of small targets are ignored. For small targets, the detection efficiency is not high, and only the low-level feature semantic information is used for fine-grained feature extraction, which is relatively accurate. YOLOv3-Tiny network follows the idea of different scale predictions in YOLOv3. After multiple rounds of upsampling and fusion of features to the last layer, the features of the last layer are predicted to form an image feature pyramid structure model. YOLOv3-Tiny network is fused with $13 * 13$ small-scale YOLO layer and $26 * 26$ mesoscale layer, respectively. According to the definition of the

International Society of Optical Engineering (SPIE), the target with an imaging area less than 80 pixels in the image with a small target of 256 ∗ 256, that is, if the target size is less than 0.12% of the original image, it can be considered as a small target. In this paper, taking 416 416 images as the input, the target with an imaging area less than 200 pixels is considered as a small target. YOLOv3-Tiny network is fused with 13 ∗ 13 small-scale YOLO layer and 26 ∗ 26 mesoscale layer, respectively, and expands 52 ∗ 52 large-scale layer on the original basis for multi-scale feature fusion prediction.

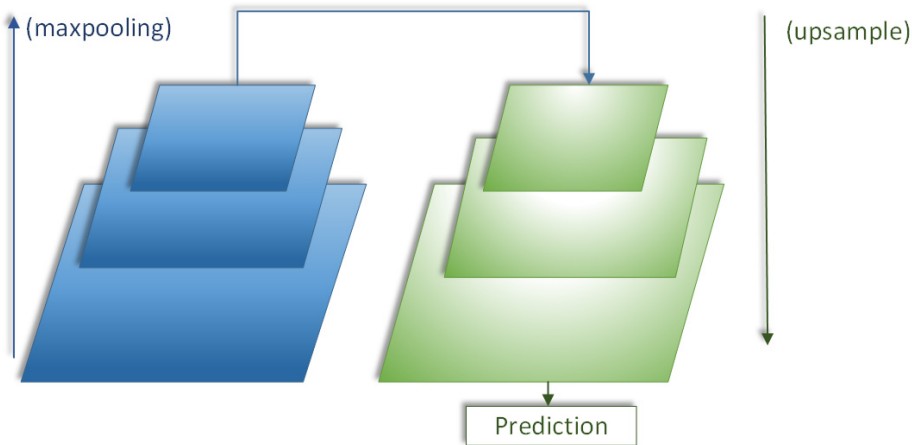

**Figure 2.** Top-level feature prediction structure.

Figure 2 shows the common structure of the original target detection algorithm. The top-level feature prediction is adopted. After multiple rounds of upsampling and fusion of features to the top-level, the top-level feature is used for prediction. After improving the YOLOv3-Tiny network, the image feature pyramid structure is realized, as shown in Figure 3. Using the multi-level and multi-scale characteristics of this structure, the visual saliency region can be accurately captured without any prior knowledge, but the rescaling operation caused by the step-by-step operation of the maximum pool layer and convolution layer will change the receptive field of the feature map, cause the loss of spatial information hierarchy, affect the accuracy of pixel-level classification and reduce the performance of the saliency prediction network.

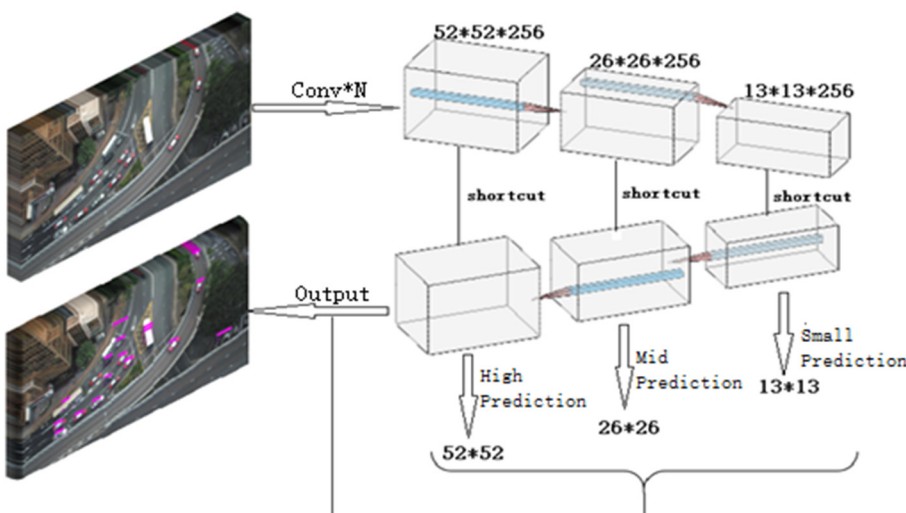

**Figure 3.** Image feature pyramid prediction structure.

### 3.2. Spatial Attention Module Is Injected Based on the Residual Structure

According to the improved YOLOv3-Tiny network based on the image feature pyramid structure, the rescaling operation caused by multi-level and multi-scale operation will change the receptive field of the feature map, cause the loss of spatial information hierarchy, affect the accuracy of pixel-level classification and reduce the performance of significance prediction network. In this regard, the YOLOv3-Tiny network adds an attention module based on the residual structure to accurately locate the target features in space.

In the target detection data set, the proportion of small-target pixels is small. Adding spatial domain attention can enhance the compactness between different regions by modeling the correlation between different regions in the image so as to enhance the characteristic response of small targets. At the same time, the channel attention module and spatial attention module are tested on the PascalVOC2007 data set. The results show that the SE module represented by the channel attention module improves the accuracy by 1% and increases the amount of calculation by 2% and speed loss by 10%, while the SAM module represented by the spatial attention module improves the accuracy by 0.5% and only increases the amount of calculation by 0.1%. The specific SAM attention module model is shown in Figure 4.

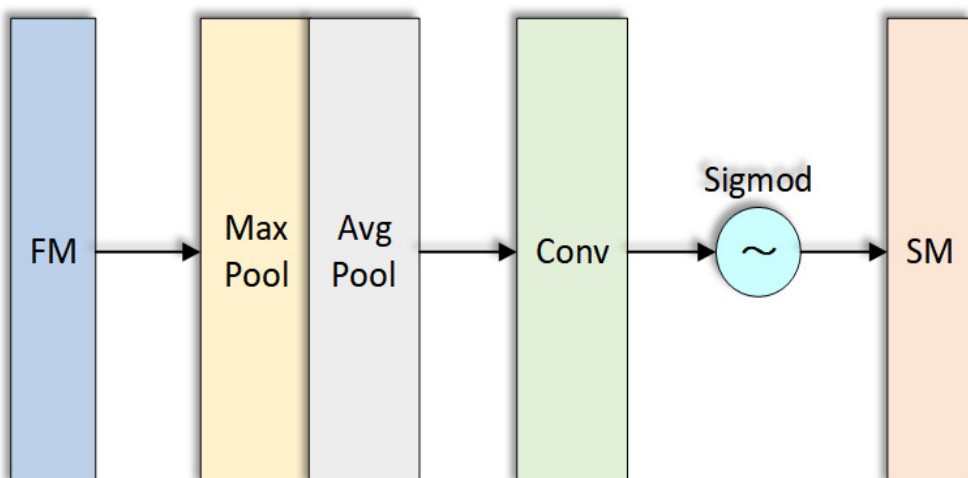

**Figure 4.** Structure of spatial attention module SAM.

Each pixel in the feature map is fed back through maximum pooling and average pooling, and the response weight of each pixel is obtained. Gradient backpropagation is carried out according to the maximum weight value, and finally, the region with the largest response in the feature map is obtained, which is regarded as the visual significance region. The time complexity of model training will affect the convergence ability of the network model. Through the residual structure, we can speed up the calculation of the weight of the spatial attention module and shorten the training time.

The spatial attention module is directly added to the original network structure vertically. The addition of the new module will increase the network depth. The deeper the network, the more abstract the features extracted and the more semantic the information. However, simply increasing the number of network layers will cause the gradient to disappear, so the residual network is added to deepen the network horizontally, as shown in Figures 5 and 6. It can obtain more semantic information about features and solve the problem of gradient dispersion in gradient descent.

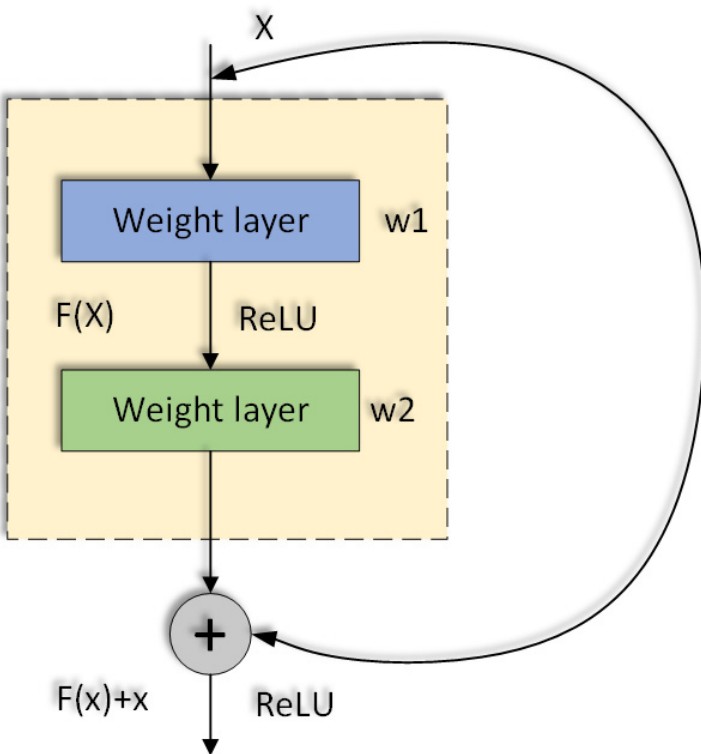

**Figure 5.** Residual block structure.

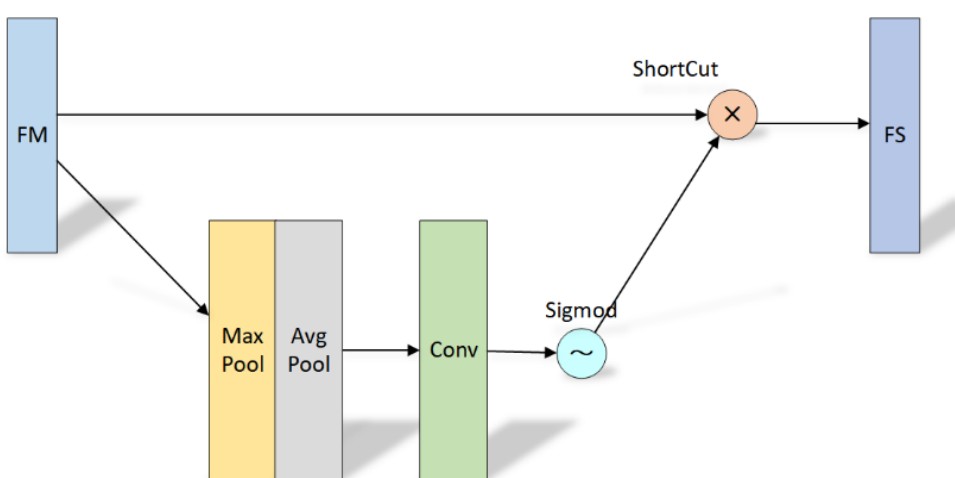

**Figure 6.** Adding SAM based on the residual structure.

When the chain derivative is used to obtain the partial derivative of a parameter, the result of the partial derivative is always small in the deep network, resulting in too-small parameter updates. Therefore, the Loss value is smaller in the process of backpropagation by calculating the gradient in the training process, which increases the time complexity of model training and affects the convergence ability of the network model. The Loss function is shown in Equation (1), and the gradient calculation is shown in Equation (2).

$$Loss = F(X_L, W_L, b_L) \tag{1}$$

$$\frac{\partial Loss}{\partial X_L} = \frac{\partial F(X_L, W_L, b_L)}{\partial X_L} \tag{2}$$

where $X_L$ represents the input value of the current layer, $W_L$ represents the weight value of the current layer and $b_L$ represents the offset term of the current layer. In the backpropagation process, the gradient of input $X_1$ is calculated as shown in Equation (3).

$$\frac{\partial X_L}{\partial X_1} = \frac{\partial F_N(X_{LN}, W_{LN}, b_{LN})}{\partial X_L} * \ldots * \frac{\partial F_2(X_{L2}, W_{L2}, b_{L2})}{\partial X_1} \tag{3}$$

By calculating the gradient, it can be seen that if one of the derivatives is very small, the gradient may become smaller and smaller after multiple multiplications, which is often called gradient dissipation. With the deepening of the network, for the deep network, the gradient value is very small when it is transferred from the deep layer close to the output to the shallow layer close to the input so that the shallow layer cannot be updated effectively. When the residual block is added, the gradient is calculated as shown in Equation (4).

$$\frac{\partial X_L}{\partial X_l} = \frac{\partial X_l + F(X_l, W_l, b_l)}{\partial X_l} = 1 + \frac{\partial F(X_L, W_L, b_L)}{\partial X_L} \tag{4}$$

At this time, the related problems of gradient descent are solved by the above methods. Moreover, after adding the fast residual, the fitting of function $F(x) = 0$ will be easier than $F(x) = x$ and the change of parameters will be more stable. When the residual learning is in the shallow layer, the learning is linear superposition, and when it reaches the deep layer, $F(x)$ will tend to zero, realizing the identity mapping so as to avoid the increase in layers affecting the learning results. At this time, on the premise of not affecting the learning results, we can increase the number of network layers, extract more fine-grained features and obtain more small-target feature information.

Considering that the model is more lightweight, the weight response can still be obtained for each pixel in the feature map by replacing the pooling operation with a $1 * 1$ convolution layer, and finally, the maximum value of regional weight can be obtained. The specific structure is shown in Figure 7.

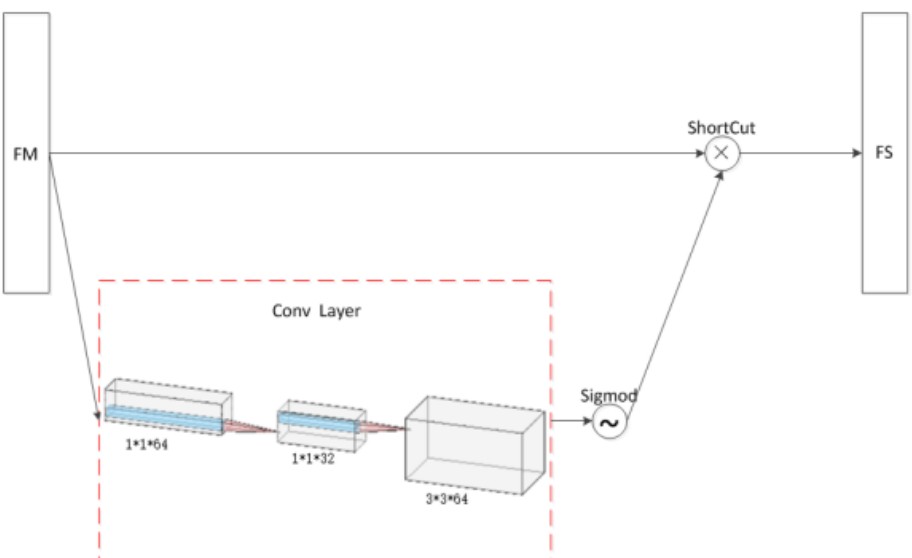

**Figure 7.** Add attention mechanism based on SAM module (the red box is the improved part).

*3.3. K-Means Algorithm Improves the Prior Size of the Target Prediction Candidate Box*

The improved YOLOv3-Tiny model follows the idea of a preset anchor box, sets 9 anchor boxes as target prediction candidate boxes and clusters the target types in the data set through the clustering algorithm to obtain the scale size of the target preselection box. As a classical clustering algorithm, the k-means algorithm is widely used because of its simplicity and efficiency. The k-means algorithm is sensitive to the selection of initial seed points, and the selection of initial seed points has a great impact on the final

clustering results and running time, and completely random selection may lead to slow convergence of the algorithm. Optimized for random initialization of seed points in the k-means algorithm. The basic idea of the improved algorithm is to select points far away from each other as seed points. The steps are as follows:

- Step 1: Randomly select an object from the input object set as the first cluster center;
- Step 2: Calculate the distance between each object and the nearest cluster center;
- Step 3: Select a new object as the new cluster center. The principle of selection is that the points with large distance are more likely to be selected as the cluster center;
- Step 4: Repeat steps (2) and (3) until K clustering centers are selected;
- Step 5: Finally, the k clustering centers are used as the initial clustering centers of the standard k-means algorithm.

### 3.4. The Goal Frame Regression Strategy Model Based on the Reward Module Is Established

For windows, four-dimensional vectors (x, y, w, h) are generally used to represent the coordinates of the center point and the width and height of the window, respectively. For Figure 8, the red box P represents the original proposal candidate target box, the blue box $G'$ represents the target box predicted by the boundary box regression algorithm, the green box $G$ represents the ground truth real target box of the target, the red circle represents the center point of the selected candidate target box and the green circle represents the center point of the real target box. The blue circle represents the center point of the target box predicted by the selected bounding box regression algorithm. Our goal is to find a relationship so that the input original window P is mapped to a regression window $G$ closer to the real window $G'$.

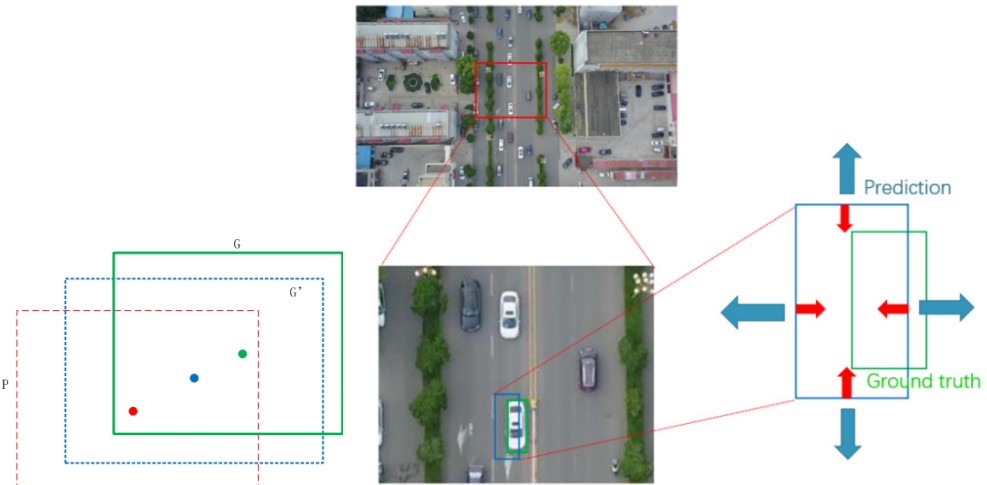

**Figure 8.** Boundary box regression(The red box P represents the original proposal candidate target box, the blue box $G'$ represents the target box predicted by the bounding box regression algorithm, the green box $G$ represents the ground truth target box of the target, the red circle represents the center point of the candidate frame, the green circle represents the center point of the real target frame, and the blue circle represents the center point of the selected bounding box regression algorithm predicts the target box).

Reinforcement learning provides a formal framework to solve the problem of what strategies agents (agent) adopt to maximize cumulative rewards in the environment. Regarding the entire image as an environment, the agent deforms the bounding box through concentrated actions. The goal is to make the bounding box tightly surround the target [21]. In order to build a complete reinforcement learning system, specific actions, states and reward functions are defined for target positioning tasks. According to the idea of reinforcement learning, the goal frame regression strategy model based on the reward module is established as follows [22].

Action set A: The set of actions an agent can take in order to achieve its goals. This article defines nine actions to search for the candidate area, including eight actions to deform the candidate area and one action to terminate the search. The specific definition of the action set is as follows: A: {Rightward, leftward, up, down, bigger, smaller, wider, higher, stop}. Each action is proportional to the size of the current bounding box $\alpha$. If a discrete change to its size is made, the terminated action represents that the agent thinks it has found the target.

Status set s: It represents the agent's understanding of the information of the current environment. The state is defined as a tuple: $s = (o, h)$. $O$ is a feature vector of the current observation area, that is, the four-dimensional vector of the target frame, which is composed of the YOLOv3-Tiny_ReSAM network is used to extract, and $h$ is a fixed size vector, representing the action history taken by the agent, and the output is a vector with dimension 9, representing nine actions.

Reward function $R$: It represents the judgment of the environment on the quality of the selected action in this state and is used to guide the agent to learn the optimal strategy according to the current state. When the agent takes action $a$, when entering the next state $s'$ from state $s$, a reward $R_a(a, s \rightarrow s')$ is given by the environment to agent. The reward signal defines what actions are close to the target and what actions are far away from the target for the agent in the current state, that is, whether the actions taken in the current state contribute to the positioning of the target. The definition of the reward function is as follows.

$$R_a(a, s \rightarrow s') = sign(DIoU(b', g, c) - DIoU(b, g, c)) \tag{5}$$

where $DIoU$ is the intersection deformation ratio $IoU$ between target $g$ and boundary box $b$:

$$DIoU(b, g, c) = IoU(b, g) - \frac{\rho^2(b, g)}{c^2} \tag{6}$$

where the center points of the prediction frame and the real frame are denoted by $b$ and $g$ and the Euclidean distance between the two center points is denoted by $\rho$. $c$ is the diagonal distance of the minimum closure region that can contain both the prediction frame and the real frame. According to the action set, state set and reward function, the Berman equation is used to iteratively update the function:

$$Q(s, a) = r + \gamma \max_a Q(s', a') \tag{7}$$

where $s$ is the current state, $a$ is the currently selected action, $R$ is the immediate reward, $\gamma$ represents the discount coefficient, $s'$ represents the next state and $a'$ represents the next action. After several iterations, the action is terminated. Finally, the experimental results will be obtained by comparing the accuracy of the target frame established by the model with that of the original algorithm.

## 4. Experimental Setup

### 4.1. Test the Target Detection Accuracy of the Improved Network Structure

Based on the above, YOLOv3-Tiny_ReSAM network structure and the original YOLOv3-Tiny network structure are tested against the open-source data set of VisionDrone2019 UAV aerial photography, and the following test results are obtained. After the improvement, the loss value for training 20,000 iterations is reduced to 18 and the loss value for training 100,000 epochs iterations is reduced to 30, as shown in Figure 9.

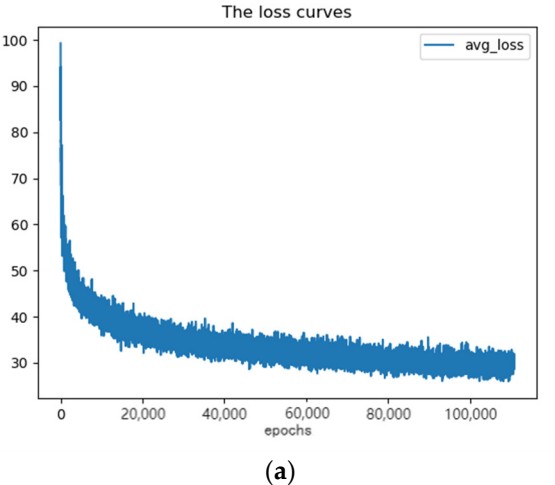
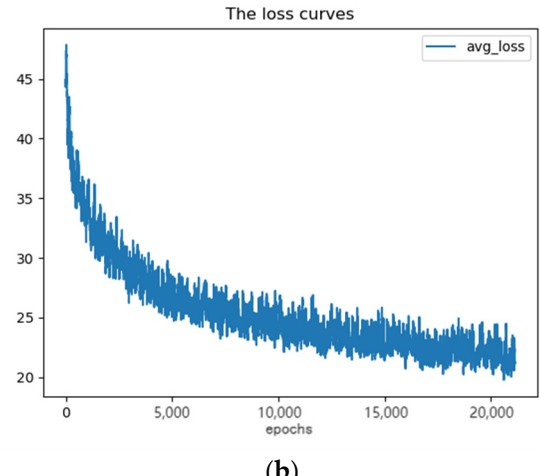

**(a)**                                                                 **(b)**

**Figure 9.** Loss curves during training: (**a**) Loss value for training 100,000 iterations. (**b**) Loss value for training 20,000 iterations.

The test set of VisionDrone2019 UAV aerial photography open source data set is tested and the recall rate results are obtained. The recall rate reflects the number ratio of various targets successfully detected in the whole data test set. The test results of 300 iterations are drawn as shown in Figure 10. The results show that the missed detection rate of the improved network model for various targets decreases significantly. With the increase in the number of iterations, according to the curve trend, the recall rate increases and the missed detection rate decreases.

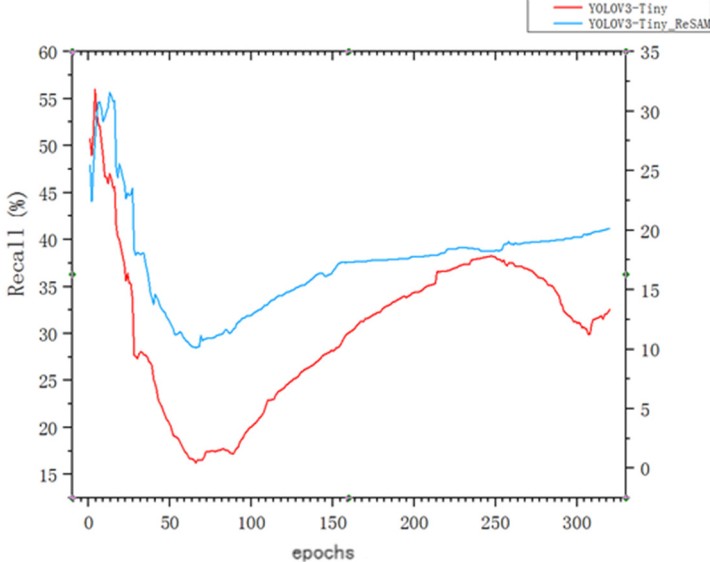

**Figure 10.** Recall rate curve.

The open-source data set of VisionDrone2019 UAV aerial photography is tested and the average accuracy results are obtained. The average accuracy reflects the number ratio of various targets correctly identified in the whole data test set. By comparing two different network model tests, the average accuracy (AP) and mean average precision(mAP) of various targets are obtained, as shown in Table 1. Among them, the mAP0.5 value of the test result obtained by the YOLOv3-Tiny network model due to the small structure of the model is low and the mAP0.5 value of the YOLOv3_ReSAM network model test result is in the middle position, which is slightly lower than the mAP0.5 value of the YOLOv4 network model and the improved TPH_YOLOv5 network model. The improved

YOLOv3_ReSAM network model has poor detection effect in two types of target detection, pedestrian and people, and has a significant advantage in the four types of targets of car, bus, motor and tricycle. Table 2 shows the results of the VisionDrone2019-Challenge competition. The detection model proposed in this paper is only 4.22% lower than the first mAP, as shown in Table 2.

**Table 1.** Average precision of various targets under different network models (Bold is an improved model based on YOLOv3-Tiny).

| | VisDrone2019-DET-Test | | | | | | | | | | mAP |
|---|---|---|---|---|---|---|---|---|---|---|---|
| | Pedestrian | People | Bicycle | Car | Van | Trunk | Tricycle | Awing-Tricycle | Bus | Motor | |
| YOLOv3_Tiny | 15.52% | 15.66% | 25.19% | 80.21% | 43.83% | 25.64% | 16.75% | 14.22% | 34.85% | 38.96% | 22.08% |
| **YOLOv3_ReSAM** | **17.28%** | **15.78%** | **28.70%** | **84.31%** | **53.77%** | **42.96%** | **30.48%** | **24.38%** | **62.35%** | **61.57%** | **33.15%** |
| YOLOv4 [23] | 44.03% | 28.18% | 23.14% | 84.67% | 52.67% | 54.31% | 31.416% | 26.72% | 65.36% | 45.87% | 35.72% |
| TPH_YOLOv5 [24] | 29% | 16.75% | 15.69% | 68.94% | 49.79% | 45.16% | 27.33% | 24.72% | 61.80% | 30.90% | 37.32% |

**Table 2.** VisionDrone2019-Challenge competition results(The first was the model that won first place in the VisionDrone2019-Challenge competition, the eighth is an improved model based on YOLOv3-Tiny).

| Ranking | Model and Method | mAP (%) |
|---|---|---|
| 1 | DPNetV3 | 37.37 |
| 2 | SMPNet | 35.98 |
| 3 | DBNet | 35.73 |
| 4 | CDNet | 34.19 |
| 5 | ECascade R-CNN | 34.09 |
| 6 | FPAFS-CenterNet | 32.34 |
| 7 | DOHR-RetinalNet | 21.68 |
| 8 | YOLOv3-Tiny-ReSAM | 33.15% |

The accuracy of various targets is further analyzed according to Table 1 to obtain the recognition accuracy of each target in the training process, as shown in Figure 11. The improved network model has a fast convergence speed and high accuracy.

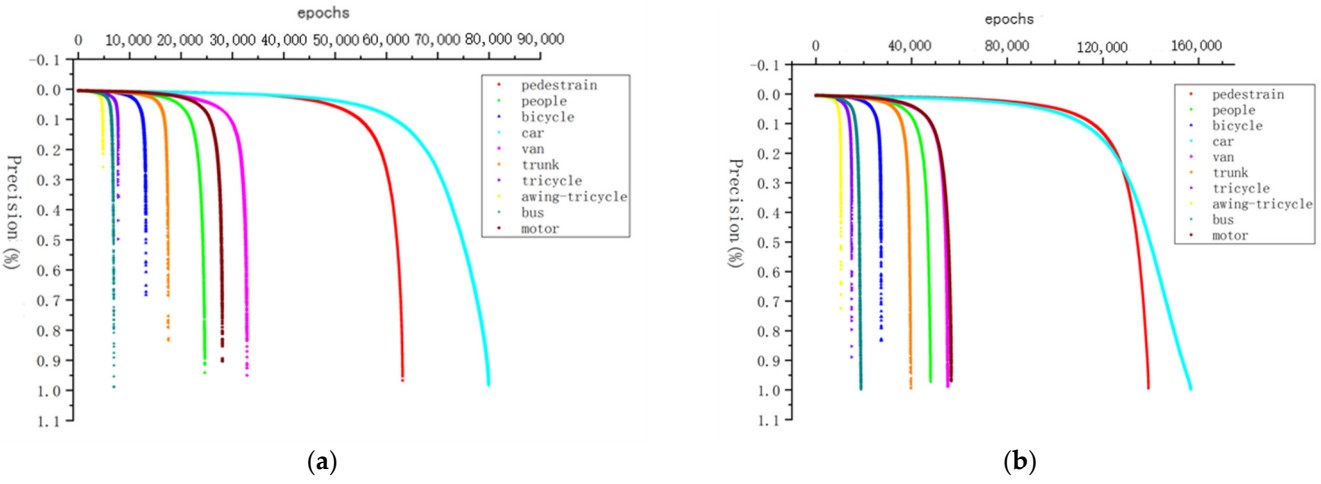

**(a)** **(b)**

**Figure 11.** (**a**) The result curve of various target recognition accuracy values under YOLOv3-Tiny network model. (**b**) The right figure is the result curve of various target recognition accuracy values under YOLOv3-Tiny_ReSAM network model.

The open-source test set images of VisionDrone2019 UAV aerial photography are selected for a visual test and the following test results are obtained. See Figure 12 for details.

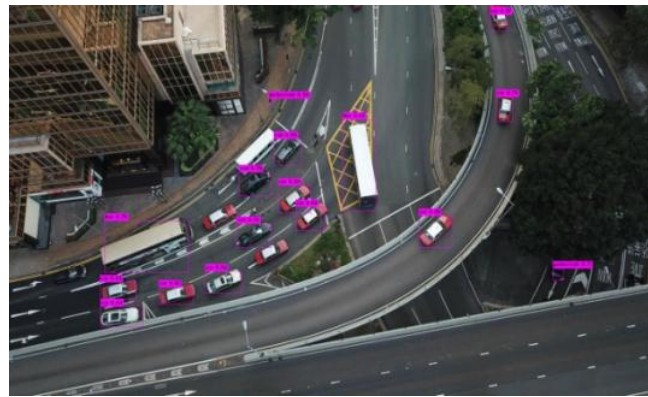 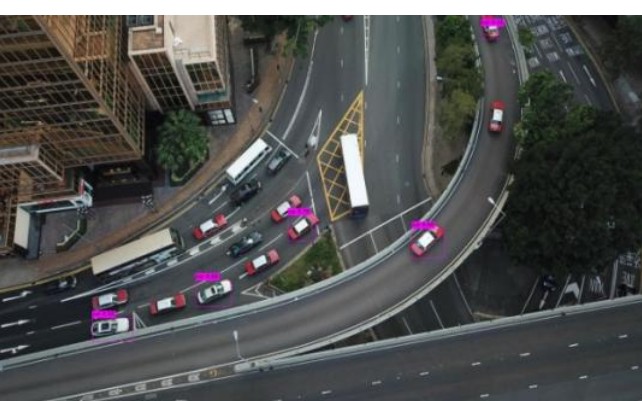

**Figure 12.** Visual test of small-target detection.

*4.2. Improved K-Means Clustering Algorithm and Gave a Priori Box Size Resultse*

By default, the YOLOv3 algorithm is set to 3 anchor boxes per unit. A larger number of anchor boxes can better cover the target, that is, the average IOU of each anchor box and its assigned target real boundary box is larger. Therefore, 9 cluster centers are set, corresponding to 3 anchor boxes per unit. Based on the open-source data set of aerial photography of VisionDrone2019, the following test results are obtained through the k-means clustering model, as shown in Table 3.

**Table 3.** Sizes of multi-scale target candidate boxes.

| Serial Number | 13 * 13 Prediction Frame (w, h) | | | 26 * 26 Prediction Frame (w, h) | | | 52 * 52 Prediction Frame (w, h) | | | Accuracy |
|---|---|---|---|---|---|---|---|---|---|---|
| 1 | 2, 3 | 4, 12 | 3, 7 | 21, 18 | 13, 18 | 12, 11 | 33, 43 | 27, 18 | 26, 26 | 31.24% |
| 2 | 1, 3 | 4, 5 | 2, 6 | 8, 9 | 3, 11 | 6, 16 | 17, 15 | 12, 26 | 30, 38 | 39.79% |
| 3 | 2, 9 | 6, 5 | 11, 10 | 13, 28 | 33, 26 | 45, 12 | 61, 34 | 58, 109 | 112, 103 | 58.63% |
| 4 | 1, 3 | 2, 8 | 4, 5 | 7, 18 | 17, 15 | 4.12 | 13.28 | 31.39 | 78, 55 | 48.51% |
| 5 | 5, 7 | 10, 11 | 12, 13 | 20, 10 | 14, 18 | 22, 11 | 56, 33 | 45, 67 | 106, 137 | 60.99% |
| <span style="color:red">6</span> | <span style="color:red">7, 18</span> | <span style="color:red">12, 11</span> | <span style="color:red">17, 15</span> | <span style="color:red">13, 29</span> | <span style="color:red">21, 18</span> | <span style="color:red">33, 43</span> | <span style="color:red">72, 63</span> | <span style="color:red">52, 121</span> | <span style="color:red">116, 124</span> | <span style="color:red">67.88%</span> |
| 7 | 1, 4 | 3, 6 | 6, 7 | 4, 12 | 7, 18 | 12, 11 | 21, 18 | 13, 30 | 33, 42 | 30.08% |
| 8 | 2, 4 | 6, 8 | 9, 7 | 11, 10 | 16, 15 | 29, 16 | 30, 49 | 40, 56 | 83, 64 | 55.39% |
| 9 | 6, 5 | 13, 9 | 10, 15 | 21, 13 | 19, 33 | 34, 15 | 74, 55 | 29, 89 | 124, 135 | 60.78% |
| 10 | 7, 3 | 13, 14 | 12, 11 | 13, 11 | 16, 23 | 33, 41 | 71, 68 | 73, 75 | 127, 118 | 67.63% |
| 11 | 6, 10 | 11, 12 | 15, 15 | 16, 12 | 20, 34 | 36, 39 | 79, 54 | 59, 88 | 121, 133 | 65.24% |
| 12 | 2, 4 | 10, 5 | 9, 8 | 14, 12 | 23, 19 | 39, 24 | 45, 57 | 67, 81 | 113, 97 | 61.27% |
| 13 | 2, 3 | 4, 7 | 9, 9 | 10, 15 | 17, 14 | 18, 26 | 30, 21 | 37, 43 | 86, 43 | 43.45% |
| 14 | 2.8 | 6.6 | 10.9 | 15.13 | 20.11 | 30.24 | 33.41 | 54, 49 | 76, 51 | 49.64% |
| <span style="color:blue">15</span> | <span style="color:blue">10, 13</span> | <span style="color:blue">16, 30</span> | <span style="color:blue">33, 23</span> | <span style="color:blue">30, 61</span> | <span style="color:blue">62, 45</span> | <span style="color:blue">59, 119</span> | <span style="color:blue">116, 90</span> | <span style="color:blue">156, 198</span> | <span style="color:blue">373, 326</span> | <span style="color:blue">43.64%</span> |

Note: Among them, the marked red is the target candidate box size with the highest accuracy under the VisionDrone2019 UAV aerial photography open-source data set and the marked blue is the default target candidate box size of the native network under the VisionDrone2019 UAV aerial photography open-source data set.

*4.3. Improved Boundary Regression Algorithm and Gave Accuracy Test Results*

The boundary regression strategy based on the reward module tests the accuracy of the boundary box. The comparison curve of the iterative test IoU of the boundary

regression algorithm before and after the improvement is shown in Figure 13. According to the curve, it can be clearly observed that under the improved boundary regression strategy, the average IOU of the boundary box accuracy (taking IoU value as the criterion) is increased from 0.485 to 0.597.

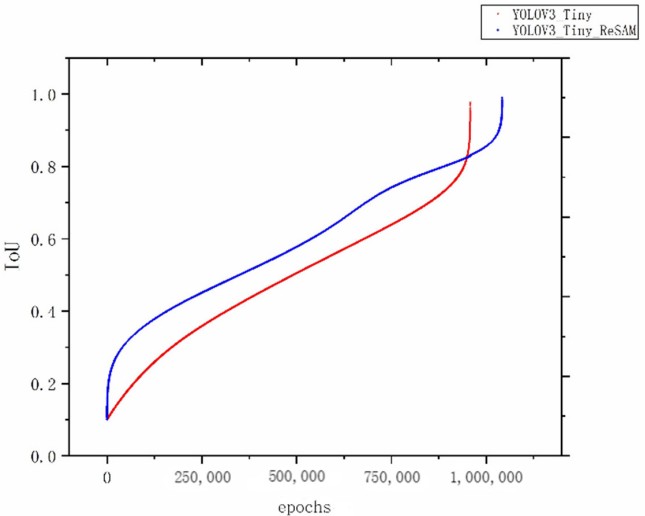

**Figure 13.** IoU curve.

The results shown in Figure 14 show that the default boundary regression algorithm of native YOLOv3 depends on the size of the boundary prediction candidate box and has poor migration ability for small targets, as shown in Figure 14a,c. On the basis of the original algorithm, fine adjustment is made on the basis of the original coarse positioning through the boundary regression strategy based on the reward module. The specific test results are shown in Figure 14b,d. Note that Figure 14 is an enlarged image in the same scale after cutting the original test image after the experiment.

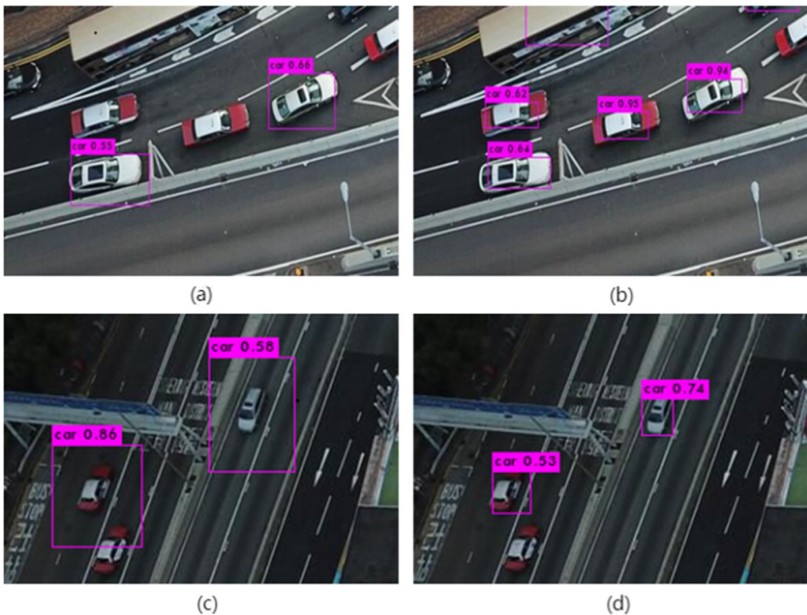

**Figure 14.** Boundary regression visualization test.

## 5. Conclusions

In this paper, the parallel structure of the feature pyramid is expanded and the shallow features are integrated so as to extract more fine-grained feature information of small

targets in the shallow network model and predict them in the top-level feature map. In the multi-level parallel feature pyramid structure, the loss of spatial information hierarchy due to multi-level and multi-scale operations is compensated for the loss of accuracy of pixel-level classification by injecting spatial attention mechanism, and the response of small target features is strengthened while weakening background features. Experimental results show that the average accuracy mAP of the improved network model is 11.07% higher than that of the native network model, and the average recall rate of a single image is stable at about 45%. However, the background of low-altitude UAV detection of small targets on the ground is complex, the target features are blurred at long distances and the network model is huge and it is difficult to achieve real-time detection, which is still being continuously improved, but there is still no ideal solution.

A boundary regression strategy based on the reward mechanism is also established, and the reinforcement learning idea is introduced on the basis of the rough positioning of boundary regression in the native network model. Experimental results show that the finely adjusted boundary regression results are 23.74% higher than those of the native boundary regression algorithm. However, in the process of boundary regression, the precision and speed of fine adjustment depend to a large extent on the rough positioning basis of the native boundary regression algorithm. If the bounding box regression error is too large under the native boundary regression strategy, it will take a lot of time to adjust in the later refinement adjustment process. At the same time, the boundary regression refinement adjustment strategy leads to an increase in the amount of additional operations, which will delay the forward inference speed of the model, thereby affecting the real-time nature of the target detection algorithm. Therefore, in later research, it is necessary to optimize the native boundary regression strategy but also to dynamically optimize the reward function according to the coarse positioning accuracy of the fine adjustment strategy.

**Author Contributions:** Investigation, B.L.; Software, H.L.; Supervision, B.L.; Visualization, H.W.; Data curation, S.W. All authors have read and agreed to the published version of the manuscript.

**Funding:** Shaanxi Provincial Natural Science Basic Research Program Project (2019JM-603).

**Data Availability Statement:** The data that support the findings of this study are openly available in VisDrone-Dataset at https://github.com/VisDrone/VisDrone-Dataset (23 March 2022).

**Conflicts of Interest:** The authors declare no conflict of interest.

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
