# Peer review of "YOLOv3_ReSAM: A Small-Target Detection Method"

_electronics, doi:10.3390/electronics11101635_

Round 1
Reviewer 1 Report
The article presents a work of sure interest and well explained. I add only two notes:
- the figures must be better realized, centered (see figure 1 and 4 for example) and made more visible, the font is often too small and I would avoid Chinese ideograms (see figure 1)
- I do not see any reference to a code present online, I think that to be properly evaluated a paper of this type requires the evaluation of the code used (there may always be a bug) and especially requires that the experiment is (at least in part) quickly reproducible
Reviewer 2 Report
I can hardly see any improvement in results. it would be better if the authors provide GitHub for this and compare the results with YOLOv3 and others models. it is also obsolete how the author could claim that the model with the addition of more layers would not be computationally expensive as compared to YOLOv3. paper is written poorly. it's hard to understand the flow of information. figures are of poor quality.
Reviewer 3 Report
Allthough intresting work, the lack of precision in the presentation/layout makes it hard to review.
Some detailed comments:
- The "Related work" section is not really a related work section - more a method/theory section.
- The formating, layout and variables in text of the paper needs consible work to make the paper acceptable for publication. Now it feels as unfinished work.
- Each sentence, heading or capitation shuld start with a capital letter.
- The headings are too long, do not make sence.
- Add reference: The work in the reference below achived better detection of small objects through temporal information into the CNN model, should be included in the related work:
- Alqaysi, H. , Fedorov, I. , Qureshi, F. Z. & O'Nils, M. (2021). A temporal boosted yolo-based model for birds detection around wind farms. Journal of Imaging, vol. 7: 11
Round 2
Reviewer 2 Report
Improve the quality of the figure. it's very poor.
Reviewer 3 Report
Thanks for the revised version of the paper, it has improved a lot. However, there are still some concerns in the paper:
- In line 103-104, you state that "Yolov3 has superior performance advantages over other network structures...", without a reference to support that - and in addition is this really true? The paper should have a clear motivation why Yolov3 is chosen for this work over Yolov4 and Yolov5 - supported by references.
- Throughout the graphs, you use baches (e.g. in figures 10 and 11), isn't it epochs you mean?
- Still some presentation concerns in the paper,
- Lines 230-239, the handling of the G variable needs to be improved in the text. There are more similar representation of variables in text e.g. lines: 266, 274, 184-186. It looks like the equations/variables are written in super script mode.
Round 3
Reviewer 3 Report
Thanks for the revised version, however the think the two first points needs to be better addressed:
- Not to motivate the selection of Yolov3 at all is not a good solution, you need to describe the base for the selection of Yolov3 in your work - so a reader nows the initial assumption of the work and that the results form this work can probably be apply on later versions, but this is left for future work.
- If you mean Epochs, then you should write epochs in the paper and in the figures.
